# Capturing Semantically Meaningful Word Dependencies with an Admixture of Poisson MRFs

**David I. Inouye**      **Pradeep Ravikumar**      **Inderjit S. Dhillon**
Department of Computer Science
University of Texas at Austin
{dinouye,pradeepr,inderjit}@cs.utexas.edu

## Abstract

We develop a fast algorithm for the Admixture of Poisson MRFs (APM) topic model [1] and propose a novel metric to directly evaluate this model. The APM topic model recently introduced by Inouye et al. [1] is the first topic model that allows for word dependencies within each topic unlike in previous topic models like LDA that assume independence between words within a topic. Research in both the semantic coherence of a topic models [2, 3, 4, 5] and measures of model fitness [6] provide strong support that explicitly modeling word dependencies—as in APM—could be both semantically meaningful and essential for appropriately modeling real text data. Though APM shows significant promise for providing a better topic model, APM has a high computational complexity because $O(p^2)$ parameters must be estimated where $p$ is the number of words ([1] could only provide results for datasets with $p = 200$). In light of this, we develop a parallel alternating Newton-like algorithm for training the APM model that can handle $p = 10^4$ as an important step towards scaling to large datasets. In addition, Inouye et al. [1] only provided tentative and inconclusive results on the utility of APM. Thus, motivated by simple intuitions and previous evaluations of topic models, we propose a novel evaluation metric based on human *evocation* scores between word pairs (i.e. how much one word "brings to mind" another word [7]). We provide compelling quantitative and qualitative results on the BNC corpus that demonstrate the superiority of APM over previous topic models for identifying semantically meaningful word dependencies. (MATLAB code available at: http://bigdata.ices.utexas.edu/software/apm/)

## 1   Introduction and Related Work

In standard topic models such as LDA [8, 9], the primary representation for each topic is simply a list of top 10 or 15 words. To understand a topic, a person must manually consider many of the possible $\binom{10}{2}$ pairwise relationships as well as possibly larger $m$-wise relationships and attempt to infer abstract meaning from this list of words. Of all the $\binom{10}{2}$ pairwise relationships probably a very small number of them are direct relationships. For example, a topic with the list of words "money", "fund", "exchange" and "company" can be understood as referring to investment but this can only be inferred from a very high-level human abstraction of meaning. This problem has given rise to research on automatically labeling topics with a topic word or phrase that summarizes the topic [10, 11, 12]. [13] propose to evaluate topic models by randomly replacing a topic word with a random word and evaluating whether a human can identify the intruding word. The intuition for this metric is that the top words of a good topic will be related, and therefore, a person will be able to easily identify the word that does not have any relationship to the other words. [2, 3, 5] compute statistics related to Pointwise Mutual Information for all pairs of top words in a topic and attempt to correlate this with human judgments. All of these metrics suggest that capturing

semantically meaningful relationships between pairs of words is fundamental to the interpretability and usefulness of topic models as a document summarization and exploration tool.

In light of these metrics, [1] recently proposed a topic model called Admixture of Poisson MRFs (APM) that relaxes the independence assumption for the topic distributions and explicitly models word dependencies. This can be motivated in part by [6] who investigated whether the Multinomial (i.e. independent) assumption of word-topic distributions actually fits real-world text data. Somewhat unsurprisingly, [6] found that the Multinomial assumption was often violated and thus gives evidence that models with word dependencies—such as APM—may be a fundamentally more appropriate model for text data.

Previous research in topic modeling has implicitly uncovered this issue with model misfit by finding that models with 50, 100 or even 500 topics tend to perform better on semantic coherence experiments than smaller models with only 10 or 20 topics [4]. Though using more topics may allow topic models to ignore the issue of word dependencies, using more topics can make the coherence of a topic model more difficult as suggested by [4] who found that using 100 or 500 topics did not significantly improve the coherence results over 50 topics. Intuitively, a topic model with a much smaller number of topics (e.g. 5 or 10) is easier to comprehend. For instance, if training on newspaper text, the number of topics could roughly correspond to the number of sections in a newspaper such as news, weather and sports. Or, if modeling an encyclopedia, the top-level topics could be art, history, science, and society. Thus, rather than using more topics, APM opens the way for a promising topic model that can overcome this model misfit issue while only using a small number of topics.

Even though APM shows promise for being a significantly more powerful and more realistic topic model than previous models, the original paper acknowledged the significant computational complexity. Instead of needing to fit $O(k(n + p))$ parameters, APM needs to estimate $O(k(n + p^2))$ parameters. [1] suggested that by using a sparsity prior (i.e. $\ell_1$ regularization of the likelihood), this computational complexity could be reduced. However, [1] could only produce some quantitative results on a very small dataset with only 200 words. In addition, the quantitative results from [1] were tentative and inconclusive on whether APM could actually perform better than LDA in coherence experiments.

Therefore, in this paper, we seek to answer two major open questions regarding APM: 1) Is there an algorithm that can overcome the computational complexity of APM and handle real-world datasets? 2) Does the APM model actually capture more semantically interesting concepts that were not possible with previous topic models? We answer the first question by developing a parallel alternating algorithm whose independent subproblems are solved using a Newton-like algorithm similar to the algorithms developed for sparse inverse covariance estimation [14]. As in [14], this new APM algorithm exploits the sparsity of the solution to significantly reduce the computational time for computing the approximate Newton direction. However, unlike [14], the APM model is solving for $k$ Poisson MRFs simultaneously whereas [14] is only solving for a single Gaussian MRF. Another difference from [14] is that the whole algorithm can be easily parallelized up to $\min(n, p)$.

For the second question about the semantic utility of APM, we develop a novel evaluation metric that more directly evaluates the APM model against human judgments of semantic relatedness—a notion called *evocation* introduced by [7]. Intuitively, the idea is that humans seek to understand traditional topic models by looking at the list of top words. They will implicitly attempt to find how these words are related and extract some more abstract meaning that generalizes the set of words. Thus, this evaluation metric attempts to explicitly score how well pairs of words capture some semantically meaningful word dependency. Previous research has evaluated topic models using word similarity measures [4]. However, our work is different from [4] in three significant ways: 1) our metrics use evocation rather than similarity (e.g. antonyms should have high evocation but low similarity), 2) we evaluate top individual word pairs instead of rough aggregate statistics, and 3) we evaluate a topic model that directly captures word dependencies (i.e. APM). We demonstrate that APM substantially outperforms other topic models in both quantitative and qualitative ways.

## 2    Background on Admixture of Poisson MRFs (APM)

**Admixtures**    The general notion of *admixtures* introduced by [1] generalizes many previous topic models including PLSA [15], LDA [8], and the Spherical Admixture Model (SAM) [16]. *Admix-*

*tures* have also been known as *mixed membership models* [17]. In contrast to mixture distributions which assume that each observation is drawn from 1 of $k$ component distributions, admixture distributions assume that each observation is drawn from an admixed distribution whose parameters are a mixture of component parameters. As examples of admixtures, PLSA and LDA are admixtures of Multinomials whereas SAM is an admixture of Von-Mises Fisher distributions. In addition, because of the connections between Poissons and Multinomials, PLSA and LDA can be seen as admixtures of independent Poisson distributions [1].

**Poisson MRFs (PMRF)**   Yang et al. [18] introduced a multivariate generalization of the Poisson that assumes that the conditional distributions are univariate Poisson which is similar to a Gaussian MRF whose conditionals are Gaussian (unlike a Guassian MRF, however, the marginals are not univariate Poisson). A PMRF can be parameterized by a node vector $\boldsymbol{\theta}$ and an edge matrix $\Theta$ whose non-zeros encode the direct dependencies between words: $\Pr_{\mathrm{PMRF}}(\boldsymbol{x}\,|\,\boldsymbol{\theta},\Theta) = \exp\big(\boldsymbol{\theta}^T\boldsymbol{x}+\boldsymbol{x}^T\Theta\boldsymbol{x}-\sum_{s=1}^{p}\ln(x_s!) - \mathrm{A}\left(\boldsymbol{\theta},\Theta\right)\big)$, where $\mathrm{A}\left(\boldsymbol{\theta},\Theta\right)$ is the log partition function needed for normalization. This formulation needs to be slightly modified to allow for positive edges using the ideas from [19]. The log partition function can be approximated by using the pseudo log-likelihood instead of the true likelihood, which means that $\mathrm{A}\left(\boldsymbol{\theta},\Theta\right) \approx \sum_{s=1}^{p}\exp(\theta_s + \boldsymbol{x}^T\Theta_s)$. The reader should note that because this is an MRF distribution, all the properties of MRFs apply to PMRFs including that a word is independent of all other words given the value of its neighbors. For example, in a chain graph, all the variables are correlated with each other but they have a much simpler dependency structure that can be encoded with $O(n)$ parameters. Therefore, PMRFs more directly and succinctly capture the dependencies between words as opposed to other simple statistics such as covariance or pointwise mutual information.

**Admixture of Poisson MRFs (APM)**   Inouye et al. [1] essentially constructed a new admixture model by using Poisson MRFs as the topic-word distributions instead of the usual Multinomial as in LDA. This allows for word dependencies within each topic. For example, if the word "classification" appears in a document, "supervised" is more likely to appear than in general documents. Given the admixture weights vector for a document the likelihood of a document is simply: $\Pr_{\mathrm{APM}}(\boldsymbol{x}\,|\,\boldsymbol{w},\boldsymbol{\theta}^{1\ldots k},\Theta^{1\ldots k}) = \Pr_{\mathrm{PMRF}}\big(\boldsymbol{x}\,|\,\boldsymbol{\theta}=\sum_{j=1}^{k}w_j\boldsymbol{\theta}^j,\Theta=\sum_{j=1}^{k}w_j\Theta^j\big)$ (please see Appendix A for notational conventions used throughout the paper). Inouye et al. [1] define a Dirichlet($\alpha$) prior on the admixture weights and a conjugate prior with hyperparameter $\beta$ on the PMRF parameters which can be easily incorporated as pseudo counts. For our experiments as described in Sec. 4.1, we set $\alpha = 1$ (i.e. a uniform prior on admixture weights) and $\beta = \{0,1\}$.

## 3   Parallel Alternating Newton-like Algorithm for APM

In the original APM paper [1], parameters were estimated by maximizing the joint approximate posterior over all variables.[1] Instead of maximizing jointly over all parameters, we split the problem into alternating convex optimization problems. Let us denote the likelihood part (i.e. the smooth part) of the optimization function as $g(\mathrm{W},\boldsymbol{\theta}^{1\ldots k},\Theta^{1\ldots k})$ and the non-smooth $\ell_1$ regularization term as $h$ where the full negative posterior is defined as $f = g + h$. The smooth part of the approximate posterior can be written as:

$$g = -\frac{1}{n}\sum_{i=1}^{n}\sum_{s=1}^{p}\Big[\sum_{j=1}^{k}w_{ij}x_{is}(\theta_s^j + \boldsymbol{x}_i^T\Theta_s^j) - \exp\big(\sum_{j=1}^{k}w_{ij}(\theta_s^j + \boldsymbol{x}_i^T\Theta_s^j)\big)\Big], \qquad (1)$$

where $\boldsymbol{x}_i$ is the word-count vector for the $i$th document, $\boldsymbol{w}_i$ is the admixture weight vector for the $i$th document, and $\boldsymbol{\theta}^j$ and $\Theta^j$ are the PMRF parameters for the $j$th component (see Appendix B for derivation). By writing $g$ in this form, it is straightforward to see that even though the whole optimization problem is not convex because of the interaction between the admixture weights $w$ and the PMRF parameters, the problem is convex if either the admixture weights W or the component parameters $\boldsymbol{\theta}^{1\ldots k},\Theta^{1\ldots k}$ are held fixed. To simplify the notation in the following sections, we combine

the node (which is analogous to an intercept term in regression) and edge parameters by defining $\boldsymbol{z}_i = [1 \ \boldsymbol{x}_i^T]^T$, $\boldsymbol{\phi}_s^j = [\theta_s^j \ (\Theta_s^j)^T]^T$ and $\boldsymbol{\Phi}^s = [\boldsymbol{\phi}_s^1 \ \boldsymbol{\phi}_s^2 \cdots \boldsymbol{\phi}_s^k]$.

Thus, we can alternate between optimizing two similar optimization problems where one has a non-smooth $\ell_1$ regularization and the other has the constraint that $\boldsymbol{w}_i$ must lie on the simplex $\Delta^k$:

$$\underset{\boldsymbol{\Phi}^1, \boldsymbol{\Phi}^2, \cdots, \boldsymbol{\Phi}^p}{\arg\min} \quad -\frac{1}{n} \sum_{s=1}^{p} \left[ \text{tr}(\Psi^s \boldsymbol{\Phi}^s) - \sum_{i=1}^{n} \exp(\boldsymbol{z}_i^T \boldsymbol{\Phi}^s \boldsymbol{w}_i) \right] + \sum_{s=1}^{p} \lambda \| \text{vec}(\boldsymbol{\Phi}^s)_{\backslash 1} \|_1 \quad (2)$$

$$\underset{\boldsymbol{w}_1, \boldsymbol{w}_2, \cdots, \boldsymbol{w}_n \in \Delta^k}{\arg\min} \quad -\frac{1}{n} \sum_{i=1}^{n} \left[ \boldsymbol{\psi}_i^T \boldsymbol{w}_i - \sum_{s=1}^{p} \exp(\boldsymbol{z}_i^T \boldsymbol{\Phi}^s \boldsymbol{w}_i)) \right], \quad (3)$$

where $\boldsymbol{\psi}_i$ and $\Psi^s$ are constants in the optimization that can be computed from the data matrix $X$ and the other parameters that are being held fixed (see Alg. 2 in Appendix D for computation of $\Psi^s$). This alternating scheme is analogous to Alternating Least Squares (ALS) for Non-negative Matrix Factorization (NMF) [22] and EM-like algorithms such as $k$-means. By writing the optimization as in Eq. 2 and Eq. 3, we also expose the simple independence between the subproblems because they are simple summations. Thus, we can easily parallelize both optimization problems upto $\min(n, p)$ with little overhead and simple changes to the code—in our MATLAB implementation, we only changed a `for` loop to a `parfor` loop.

## 3.1 Newton-like Algorithms for Subproblems

For each of the subproblems, we develop Newton-like optimization algorithms. For the component PMRFs, we borrow several important ideas from [14] including *fixed* and *free* sets of variables for the $\ell_1$ regularized optimization problem. The overall idea is to construct a quadratic approximation around the current solution and approximately optimize this simpler function to find a step direction. Usually, finding the Newton direction requires computing the Hessian for all the optimization variables but because of the $\ell_1$ regularization, we only need to focus on variables that might be non-zero. This set of *free* variables, denoted $\mathcal{F}$, can be simply determined from the gradient and current iterate [14]. Since usually there is only a small number of *free* variables compared to *fixed* variables (i.e. $\lambda$ is large enough), we can simply run coordinate descent on these free variables and only implicitly calculate Hessian information as needed in each coordinate descent step. After finding an approximate Newton direction, we find a step size that satisfies the Armijo rule and then update the iterate (see Alg. 2 in Appendix D).

We also employed a similar Newton-like algorithm for estimating the admixture weights. Instead of the $\ell_1$ regularization term, however, this subproblem has the constraint that the admixture weights $\boldsymbol{w}_i$ must lie on the simplex so that each document can be properly interpreted as a convex mixture of over topic parameters. For this constraint, we used a dual-coordinate descent algorithm to find the approximate Newton direction as in [23].

Finally, we put both subproblem algorithms together and alternate between the two (see Alg. 1 in Appendix D). For tracing through different $\lambda$ parameters, $\lambda$ is initially set to $\infty$ so that the model trains an independent APM model first. Then, the initial $\lambda = \lambda_{\max}$ is found by computing the largest gradient of the final independent iteration. Every time the alternating algorithm converges, the value of $\lambda$ is decreased so that a set of models is trained for decreasing values of $\lambda$.

## 3.2 Timing Results

We conducted two main timing experiments to show that the algorithm can be efficiently parallelized and the algorithm can scale to reasonably large datasets. For the parallel timing experiment, we used the BNC corpus described in Sec. 4.1 ($n = 4049$, $p = 1646$) and fixed $k = 5$, $\lambda = 8$ and a total of 30 alternating iterations. For the large data experiment, we used a Wikipedia dataset formed from a recent Wikipedia dump by choosing the top 10k words neglecting stop words and then selecting the longest documents. We ran several main iterations of the algorithm with this dataset while fixing the parameters $k = 5$ and $\lambda = 0.5$. All timing experiments were conducted on the TACC Maverick system with Intel Xeon E5-2680 v2 Ivy Bridge CPUs (2.80 GHz), 20 CPUs per node, and 12.8 GB memory per CPU (https://www.tacc.utexas.edu/).

The parallel timing results can be seen in Fig. 1 (left) which shows that the algorithm does have almost linear speedup when parallelizing across multiple workers. Though we only had access to a single computer with 20 processors, substantially more speed up could be obtained by using more processors on a distributed computing system. This simple parallelism makes this algorithm viable for much larger datasets. The timing results for the Wikipedia can be seen in Fig. 1 (right). These results give an approximate computational complexity of $O(np^2)$ which show that the proposed algorithm has the potential for scaling to datasets where $n$ is $O(10^5)$ and $p$ is $O(10^4)$. The $O(p^2)$ comes from the fact that there are $p$ subproblems and each subproblem needs to calculate the gradient which is $O(p)$ as well as approximate the Newton direction for a subset of the variables. The first iteration takes longer because the initial parameter values are naïvely set to 0 whereas future iterations start from reasonable initial value.

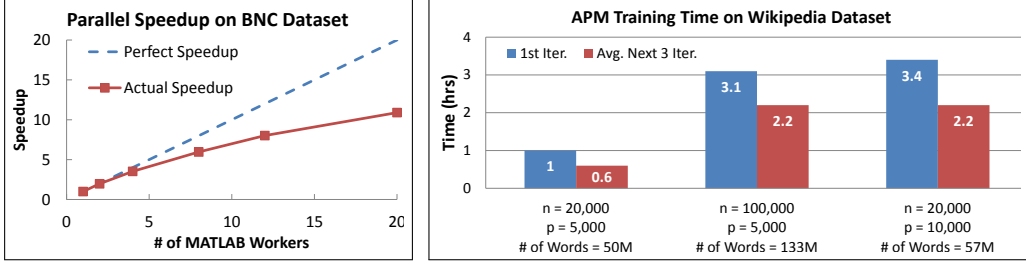

Figure 1: (left) The speedup on the BNC dataset shows that the algorithm scales approximately linearly with the number of workers because the subproblems are all independent. (right) The timing results on the Wikipedia dataset show that the algorithm scales to larger datasets and has a computational complexity of approximately $O(np^2)$.

## 4 Evocation Metric

Boyd-Graber et al. [7] introduced the notion of *evocation* which denotes the idea of which words "evoke" or "bring to mind" other words. There can be many types of evocation including the following examples from [7]: [rose - flower] (example), [brave - noble] (kind), [yell - talk] (manner), [eggs - bacon] (co-occurence), [snore - sleep] (setting), [wet - desert] (antonymy), [work - lazy] (exclusivity), and [banana - kiwi] (likeness). This is distinctive from word similarity or synonymy since two words can have very different meanings but "bring to mind" the other word (e.g. antonyms). This notion of word relatedness is a much simpler but potentially more semantically meaningful and interpretable than word similarity. For instance, "work" and "lazy" do not have similar meanings but are related through the semantic meanings of the words. Another difference is that—unlike word semantic similarity— words that generally appear in very different contexts yet mean the same thing would probably not have a high evocation score. For example, "networks" and "graphs" both have a definition that means a set of nodes and edges yet usually one word is chosen in a particular context.

Recent work in evaluating topic models [2, 3, 4, 5] formulate automated metrics based on automatically scoring all pairs of top words and noticing that they correlate with human judgment of overall topic coherence. All of these metrics are based on the common assumption that a person should be able to understand a topic by understanding the abstract semantic connections between the word pairs. Thus, *evocation* is a reasonable notion for evaluating topic modeling because it directly evaluates the level of semantic connection between word pairs. In addition, this new evocation metric provides a way to explicitly evaluate the edge matrices of APM, which would be ignored in previous metrics because explicit word dependencies are not modeled in other topic models.

We now formally define our evocation metric. Given human-evaluated scores for a subset of word pairs $\mathcal{H}$ and the corresponding weights given by a topic model for this subset of word pairs $\mathcal{M}$, let us define $\pi_{\mathcal{M}}(j)$ to be an ordering of the word pairs induced by $\mathcal{M}$ such that $\mathcal{M}_{\pi(1)} \geq \mathcal{M}_{\pi(2)} \geq \cdots \geq \mathcal{M}_{\pi(|\mathcal{H}|)}$. Then, the top-$m$ evocation metric is simply:

$$\text{Evoc}_m(\mathcal{M}, \mathcal{H}) = \sum_{j=1}^{m} \mathcal{H}_{\pi_{\mathcal{M}}(j)}. \tag{4}$$

Note that the scaling of $\mathcal{M}$ is inconsequential because $\mathcal{M}$ is only needed to define an ordering or ranking of $\mathcal{H}$. For example, $\hat{\mathcal{M}} = \alpha \exp(\mathcal{M})$ would yield the same evocation score for all scalar

values $\alpha > 0$ because the ordering would be maintained. Essentially, $\mathcal{M}$ merely induces an ordering of the word pairs and the evocation score is the sum of the human scores for these top $m$ word pairs.

For APM, the word pair weights come primarily from the PMRF edge matrices $\Theta^{1 \ldots k}$—the PMRF node vectors are only used to provide an ordering if there are not enough non-zeros in the edge matrices. For the other Multinomial-based topic models which do not have parameters explicitly associated with word-pairs, we can compute the most likely word pairs in a topic by multiplying their corresponding marginal probabilities. This weighting corresponds to the probability that two independent draws from the topic distribution produce the word pair and thus is the most obvious choice for Multinomial-based topic models.

Since this metric only gives a way to evaluate one topic, we consider two ways of determining the overall evocation score for the whole topic model: Evoc-1 $= \sum_{j=1}^{k} \frac{1}{k} \text{Evoc}_m(\mathcal{M}^j, \mathcal{H})$ and Evoc-2 $= \text{Evoc}_m(\sum_{j=1}^{k} \frac{1}{k} \mathcal{M}^j, \mathcal{H})$. In words, these are "average evocation of topics" and "evocation of average topic" respectively. Evoc-1 measures whether all or at least most topics capture meaningful word associations since it can be affected by uninteresting topics. Evoc-2 is reasonable for measuring whether the topic model as a whole is capturing word semantics even if some of the topics are not capturing interesting word associations. This second measure has some relation to the word similarity measure of topic coherence in [4]. However, [4] uses similarity rather than evocation, does not directly evaluate top individual word pairs and does not evaluate any models with word dependencies such as APM.

## 4.1 Experimental Setup

**Human-Scored Evocation Dataset**  The original human-scored evocation dataset was produced by a set of trained undergraduates in which 1,000 words were hand selected primarily based on their frequency and usage in the British National Corpus (BNC) [7]. From the possible pairwise evaluations, approximately 10% of the word pairs were randomly selected to be manually scored by a set of trained undergraduates. The second dataset was constructed by predicting the pairs of words that were likely to have a high evocation using a standard machine learning classifier. This new set of pairs was scored using Amazon MTurk (mturk.com) by using the original dataset as a control [24]. Though these scores are between synsets—which are a word, part-of-speech and sense triplet—, we mapped all of the synsets to word, part-of-speech pairs since that is the only information we have for the BNC corpus. This led to a total of 1646 words. In addition, though the evocation dataset has scores for directed relationships (i.e. word1 $\rightarrow$ word2 could have a different score than word2 $\rightarrow$ word1), we averaged these two scores because the directionality of the relationship is not modeled by APM or any other topic model.

**BNC Corpus**  Because the evocation dataset was based on the BNC corpus, we used the BNC corpus for our experiments. We processed the BNC corpus by lemmatizing each word using the Word-NetLemmatizer included in the nltk package (nltk.org) and then attaching the part-of-speech, which is already included in the BNC corpus. We only retained the counts for the 1646 words that occurred in the human-scored datasets but processed all 4049 documents in the corpus.

**APM Model Parameters**  We trained APM on the BNC corpus with several different parameter settings including various $\lambda$ and $\beta$ parameter settings. We also trained two particular APM models denoted APM-LowReg and APM-HeldOut. APM-LowReg uses a very small regularization parameter so that almost all edges are non-zero. APM-HeldOut automatically selects a reasonable value for $\lambda$ based on the likelihood of a held-out set of the documents. Thus, the APM-HeldOut model does not require a user-specified $\lambda$ parameter but—as seen in the following sections—still performs reasonably well even compared to the APM model in which many different parameter settings are attempted. In addition, the APM-HeldOut can stop the training early when the model begins to overfit the data rather than tracing through all the $\lambda$ parameters—this could lead to a significant gain in model training time. The authors suggest that APM-HeldOut is a simple baseline model for future comparison if a user does not want to specify $\lambda$.

**Other Models**  For comparison, we trained five other models: Correlated Topic Models (CTM), Hierarchical Dirichlet Process (HDP), Latent Dirichlet Allocation (LDA), Replicated Softmax (RSM), and a naïve random baseline (RND). CTM models correlations between topics [25]. HDP

is a non-parametric Bayesian model that selects the number of topics based input data and hyperparameters [26]. The standard topic model LDA was trained using MALLET [27]. LDA was trained for at least 5,000 iterations and HDP was trained for at least 300 iterations since HDP is computationally expensive. RSM is an undirected topic model based on Restricted Boltzmann Machines (RBM) [28]. The random model is merely the expected evocation score if edges are ranked at random. We ran a full factorial experimental setting where all the combinations of a set of parameter values were trained to give a fair comparison between models (see Appendix C for a summary of parameter values). All these comparison models only indirectly model dependencies between words through the latent variables since the topic distributions are Multinomials whereas APM can directly model the dependencies between words since the topic distributions are Poisson MRFs.

**Selecting Best Parameters**  We randomly split the human scores into a 50% tuning split and 50% testing split. Note that we have a *tuning* split rather than a training split because the model training algorithms are unsupervised (i.e. they never see the human scores) so the only supervision occurs in selecting the final model parameters (i.e. during the tuning phase). Therefore, we selected the final parameters based on the tuning split and computed the final evocation scores on the test split. Thus, even when selecting the parameter settings, the modeling process never sees the test data.

## 4.2  Main Results

The Evoc-1 and Evoc-2 scores with $m = 50$ for all models can be seen in Fig. 2.[2] For Evoc-1, the APM models significantly outperform all other models for a small number of topics and even capture many semantically meaningful word pairs with a single topic. For higher number of topics, the APM models seem to perform only competitively with previous topic models. It seems that APM-LowReg performs better with a small number of topics whereas APM-HeldOut—which generally chooses a relatively high $\lambda$—seems more robust for large number of topics. These trends are likely caused because there is a relatively small number of documents ($n = 4049$) so the APM-LowReg begins to significantly overfit the data as the number of topics increases whereas APM-HeldOut does not seem to overfit as much. For all the APM models, the degradation in performance as the number of topics increases is most likely caused by the fact that a Poisson MRF with $O(p^2)$ parameters is a much more flexible distribution than a Multinomial, and thus, fewer topics are needed to appropriately model the data. These results also give some evidence that APM can succinctly model the data with a much smaller number of topics than is needed for independent topic models; this succinctness could be particularly helpful for the interpretability and intuitions of topic models.

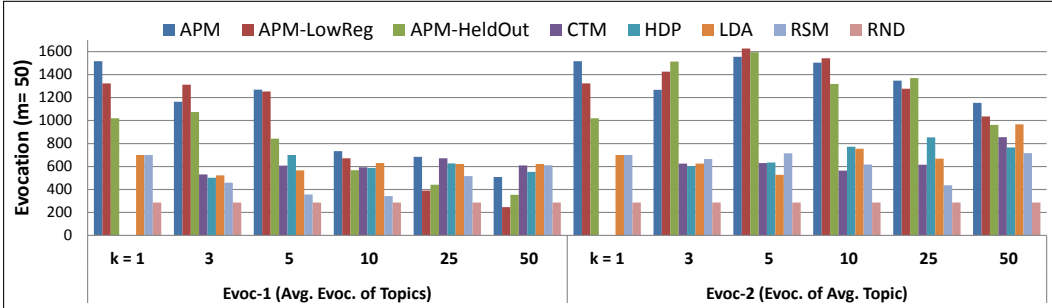

Figure 2: Both Evoc-1 scores (left) and Evoc-2 scores (right) demonstrate that APM usually significantly outperforms other topic models in capturing meaningful word pairs.

For the Evoc-2 score, the APM models—including the APM-HeldOut model which automatically determines a $\lambda$ from the data—significantly outperform previous topic models even for a large number of topics. This supports the idea that APM only needs a small number of topics to capture many of the semantically meaningful word dependencies. Thus, when increasing the number of topics beyond 5, the performance does not decrease as in Evoc-1. It is likely that this discrepancy is caused by the fact that many of the edges are concentrated in a small number of topics even when the number of topics is 10 or 25. As expected because of previous research in topic models, most other topic

models perform slightly better with a larger number of topics. Though it is possible that using 100 or 500 topics for these topic models might give an evocation score better than APM with 5 topics, this would only enforce the idea that APM can perform better or at least competitively with previous topic models while only using a comparatively small number of topics.

**Qualitative Analysis of Top 20 Word Pairs for Best LDA and APM Models**    To validate the intuition of using evocation as an human-standard evaluation metric, we present the top 20 word pairs for the best standard topic model—in this case LDA—and the best APM model for the Evoc-2 metric as seen in Table 1. The best performing LDA model was trained with 50 topics, $\alpha = 1$ and $\beta = 0.0001$. The best APM model was the APM-LowReg model trained with only 5 topics and a small regularization parameter $\lambda = 0.05$. It is important to note that the best model for LDA has 50 topics while the best model for APM only has 5 topics. As before, this reinforces the theme that APM can capture more semantically meaningful word pairs with a smaller number of topics than previous topic models.

Table 1: Top 20 words for LDA (left) and APM (right)

| Rank | Evoc. | Edge | Rank | Evoc. | Edge | Rank | Evoc. | Edge | Rank | Evoc. | Edge |
|---|---|---|---|---|---|---|---|---|---|---|---|
| 1 | 38 | woman.n ↔ man.n | 11 | 0 | car.n ↔ bus.n | 1 | 13 | smoke.v ↔ cigarette.n | 11 | 72 | aunt.n ↔ uncle.n |
| 2 | 0 | woman.n ↔ wife.n | 12 | 31 | year.n ↔ day.n | 2 | 60 | love.v ↔ love.n | 12 | 28 | tea.n ↔ coffee.n |
| 3 | 13 | train.n ↔ car.n | 13 | 25 | car.n ↔ seat.n | 3 | 13 | eat.v ↔ food.n | 13 | 25 | operational.a ↔ aircraft.n |
| 4 | 69 | school.n ↔ class.n | 14 | 50 | teach.v ↔ student.n | 4 | 50 | west.n ↔ east.n | 14 | 0 | competition.n ↔ compete.v |
| 5 | 0 | drive.v ↔ car.n | 15 | 0 | tell.v ↔ get.v | 5 | 38 | south.n ↔ north.n | 15 | 35 | green.n ↔ green.a |
| 6 | 82 | teach.v ↔ school.n | 16 | 38 | wife.n ↔ man.n | 6 | 75 | iron.n ↔ steel.n | 16 | 0 | fox.n ↔ animal.n |
| 7 | 38 | engine.n ↔ car.n | 17 | 100 | run.v ↔ car.n | 7 | 57 | question.n ↔ answer.n | 17 | 19 | smoke.n ↔ fire.n |
| 8 | 35 | publish.v ↔ book.n | 18 | 0 | give.v ↔ get.v | 8 | 13 | boil.v ↔ potato.n | 18 | 41 | wine.n ↔ drink.v |
| 9 | 7 | religious.a ↔ church.n | 19 | 16 | paper.n ↔ book.n | 9 | 7 | religious.a ↔ church.n | 19 | 33 | troop.n ↔ force.n |
| 10 | 38 | state.n ↔ government.n | 20 | 19 | white.a ↔ black.a | 10 | 97 | husband.n ↔ wife.n | 20 | 7 | lock.n ↔ key.n |

One interesting example is that LDA finds two word pairs [woman.n - wife.n] and [wife.n - man.n] that capture some semantic notion of marriage. However, APM directly captures this semantic meaning with [husband.n - wife.n]. APM also finds more intuitive verb-noun relationships that are closely tied semantically and portray a particular context: [smoke.v - cigarette.n], [eat.v - food.n], [boil.v - potato.n], and [drink.v - wine.n] whereas LDA tends to select less interesting verb-noun relationships such as [run.v - car.n]. In addition, APM finds multiple semantically coherent yet high level word pairs such as [iron.n - steel.n], [question.n - answer.n], and [aunt.n - uncle.n], whereas LDA finds several low-level edges such as [year.n - day.n] and [tell.v - get.v]. These overall trends become even more evident when looking at the top 50 edges as can be found in the Appendix E. Both the quantitative evaluation metrics (i.e. Evoc-1 and Evoc-2) as well as a qualitative exploration of the top word pairs give strong evidence that APM can succinctly capture both more interesting and higher-level semantic concepts through word dependencies than independent topic models.

# 5    Conclusion and Future Work

We motivated the need for more expressive topic models that consider word dependencies—such as APM—by considering previous work on topic model evaluation metrics. We overcame the significant computational barrier of APM by providing a fast alternating Newton-like algorithm which can be easily parallelized. We proposed a new evaluation metric based on human evocation scores that seeks to measure whether a model is capturing semantically meaningful word pairs. Finally, we presented compelling quantitative and qualitative measures showing the superiority of APM in capturing semantically meaningful word pairs. In addition, this metric suggests new evaluations of topic models based on evaluating top word pairs rather than top words. One drawback with the current human-scored data is that only a small portion of the word pairs have been scored. Thus, one extension is to dynamically collect more human scores as needed for evaluation. This work also opens the door for exciting new word-semantic applications for APM such as Word Sense Induction using topic models [29], keyword expansion or suggestion, document summarization, and document visualization because APM is capturing semantically meaningful relationships between words.

**Acknowledgments**

D. Inouye was supported by the NSF Graduate Research Fellowship via DGE-1110007. P. Ravikumar acknowledges support from ARO via W911NF-12-1-0390 and NSF via IIS-1149803, IIS-1447574, and DMS-1264033. I. Dhillon acknowledges support from NSF via CCF-1117055.

## Footnotes

[1]This posterior approximation was based on the pseudo-likelihood while ignoring the symmetry constraint so that nodewise regression parameters are independent. This leads to an overcomplete parameterization for APM. For an overview of composite likelihood methods, see [20]. For a comparison of pseudo-likelihood versus nodewise regressions, see [21].

[2]For simplicity and comparability, we grouped HDP into the topic number that was closest to its discovered number of topics because HDP can select a variable number of topics.

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
