[Supplementary Material]

## A  Notational Conventions

Matrices are denoted by capital letters (e.g. $X$, $\Theta$). Column vectors are denoted by lowercase bold face Roman and Greek letters (e.g. $\boldsymbol{x}$, $\boldsymbol{\theta}$). Usually, lower case letters are the columns of their upper case matrix counterparts (e.g. $\boldsymbol{x}_i$ is the $i$th column vector of $X$) except for $\boldsymbol{\theta}$ which is distinct from a column of $\Theta$. Subscripts indicate either the column of a matrix (e.g. $\Theta_s$) or a scalar value indexed on a vector (e.g. $x_{is}$, $\theta_s$). Superscripts indicate an element of a set, which can either be a set of vectors or a set of matrices (e.g. $\boldsymbol{\theta}^j \in \boldsymbol{\theta}^{1\ldots k}$, $\Theta^j \in \Theta^{1\ldots k}$, or $\boldsymbol{\Phi}^s \in \boldsymbol{\Phi}^{1\ldots p}$).

The subscript $\backslash i$ as in $\text{vec}(\boldsymbol{\Phi}^s)_{\backslash i}$ refers to the sub vector when the $i$th coordinate is made to be zero. This is important when calculating the $\ell_1$ regularization because the only the edge parameters are regularized and therefore the node parameters must be ignored.

## B  Reformulation of negative pseudo log likelihood

$$\mathcal{L} = -\frac{1}{n}\sum_{i=1}^{n}\Pr_{\text{APM}}(\boldsymbol{x}_i \mid \boldsymbol{w}_i, \boldsymbol{\theta}^{1\ldots k}, \Theta^{1\ldots k}) \tag{5}$$

$$= -\frac{1}{n}\sum_{i=1}^{n}\Pr_{\text{PMRF}}\left(\boldsymbol{x}_i \mid \boldsymbol{\theta}^i = \sum_{j=1}^{k} w_j\boldsymbol{\theta}^j, \Theta^i = \sum_{j=1}^{k} w_j\Theta^j\right) \tag{6}$$

$$= -\frac{1}{n}\sum_{i=1}^{n}\left[\left(\sum_{j=1}^{k} w_{ij}\boldsymbol{\theta}^j\right)^T\boldsymbol{x}_i + \boldsymbol{x}_i^T\left(\sum_{j=1}^{k} w_{ij}\Theta^j\right)\boldsymbol{x}_i - \sum_{s=1}^{p}\exp\left(\sum_{j=1}^{k} w_{ij}(\theta_s^j + \boldsymbol{x}_i^T\Theta_s^j)\right)\right] \tag{7}$$

$$= -\frac{1}{n}\sum_{i=1}^{n}\sum_{s=1}^{p}\left[\sum_{j=1}^{k} w_{ij}x_{is}(\theta_s^j + \boldsymbol{x}_i^T\Theta_s^j) - \exp\left(\sum_{j=1}^{k} w_{ij}(\theta_s^j + \boldsymbol{x}_i^T\Theta_s^j)\right)\right] \tag{8}$$

# C   Parameter Settings

A summary of the parameter settings for the models trained can be seen in Table 2. Experiments were run over all possible combinations of the parameters given and final parameter values determined by evocation score on 50% tuning set. Note that the output edge matrices of APM (i.e. $\Theta^{1\ldots k}$) are not symmetric because the algorithm ignores the symmetry constraint thus yielding an overcomplete representation in which two estimates of the word dependency parameters are computed. These two estimates can be combined in at least 2 ways:

1. $OR$: Assume the combined estimate is a non-zero if either estimate is non-zero (i.e. take the $OR$ of the estimated non-zero edges). Then, merely average both estimates.

2. $AND$: Assume the combined estimate is non-zero *only if both* estimates are non-zero (i.e. take the $AND$ of the estimated non-zero edges). Then, merely average the non-zero entries.

Note that if the estimator is actually recovering the true neighborhood (i.e. a variable's non-zero dependencies with other variables), then these definitions are equivalent. However, in practice, we have found that the models are quite different yet both give reasonable results. In general, we observed that $AND$ is easier to interpret and is less likely to overfit the training data than $OR$. $AND$ also has the intuitive interpretation that two words are directly dependent on one another if and only if they are useful in predicting each other (i.e. they are non-zero coefficients in the node-wise Poisson-like regression problems). This is why we chose to use $AND$ for APM-LowReg and APM-HeldOut. We suggest that in general $AND$ is probably a better post-processing step than $OR$. However, more fully studying the effects of this post-processing step could be an area of future research.

Table 2: Table of Parameter Settings for Models

| Model | Parameter settings |
|---|---|
| APM | $k \in \{1, 3, 5, 10, 25\}$ <br> Trace iteration $\in \{1, 2, \cdots, 15\}$ (i.e. different $\lambda$ values) <br> $\beta \in \{0, 0.01, 1\}$ <br> Post processing of edge set $\in \{AND, OR\}$ |
| APM-LowReg | $k \in \{1, 3, 5, 10, 25\}$ <br> $\lambda$ chosen to be very small (usually approximately $\frac{\lambda_{\max}}{2^{15}}$) <br> $\beta = 0$ <br> Post processing of edge set $\in \{AND\}$ |
| APM-HeldOut | $k \in \{1, 3, 5, 10, 25\}$ <br> Percentage of held-out documents $\in \{10\%, 20\%\}$ <br> $\lambda$ chosen by held-out training documents <br> $\beta = \{0, 0.1\}$ <br> Post processing of edge set $\in \{AND\}$ |
| CTM | $k \in \{1, 3, 5, 10, 25\}$ <br> Default parameters except for two different convergence criteria |
| HDP | Topic Dirichlet hyperparameter $\eta \in \{1, 0.01, 0.0001\}$ <br> Hyperparameter resampling $\in \{yes, no\}$ <br> Scaling for prior if hyperparameter resampling or first concentration parameter $\gamma \in \{100, 10, 1, 0.1\}$ |
| LDA | $k \in \{1, 3, 5, 10, 25, 50\}$ <br> Topic Dirichlet hyperparameter $\beta \in \{1, 0.01, 0.0001\}$ <br> Document Dirichlet hyperparameter $\alpha = \{1, 0.1, 0.01\}$ <br> Optimize hyperparameters $\in \{yes, no\}$ |
| RSM | $k \in \{1, 3, 5, 10, 25, 50\}$ <br> Learning rate $\in \{10^{-3}, 5 \times 10^{-4}, 10^{-4}, 5 \times 10^{-5}, 10^{-5}\}$ <br> Maximum iterations $\in \{10^3, 10^4\}$ |

# D Algorithms

## D.1 Main Alternating Algorithm for APM

---

**Algorithm 1:** Estimate APM parameters using an alternating scheme

---

**Input** : Data matrix $X \in \mathbb{Z}_+^{p \times n}$, number of topics $k$, prior hyperparameter $\beta \geq 0$
**Output**: Parameters $\boldsymbol{\theta}_\lambda^{1 \ldots k}$, $\Theta_\lambda^{1 \ldots k}$ and $\mathrm{W}_\lambda$ for different values of $\lambda$

1   $\mathrm{W} \leftarrow \mathrm{rand}(k, n)$
2   **for** $\lambda \leftarrow \infty, \lambda_{\max}, \frac{\lambda_{\max}}{2}, \frac{\lambda_{\max}}{4}, \frac{\lambda_{\max}}{8}, \cdots$ **do**
3     **while** *not converged* **do**
4       $[\boldsymbol{\theta}^{1 \ldots k}; \Theta^{1 \ldots k}] \leftarrow \mathrm{EstimateComponentPMRFs}(\mathrm{W}, X, \lambda, \beta)$
5       $\mathrm{W} \leftarrow \mathrm{EstimateAdmixtureWeights}(\boldsymbol{\theta}^{1 \ldots k}, \Theta^{1 \ldots k}, X)$
6     **end**
7   **end**

---

## D.2 Component PMRFs Algorithm

---

**Algorithm 2:** Estimates the $k$ node and edge parameters for word index $s$ when W is fixed

---

**Input** : Data matrix $X$, admixture weights matrix W, word index $s$, sparsity parameter $\lambda$
**Output**: Parameter $\Phi^s$

1   $\boldsymbol{Z} \leftarrow \begin{bmatrix} 1 & 1 & \cdots & 1 \\ \boldsymbol{x}_1 & \boldsymbol{x}_2 & \cdots & \boldsymbol{x}_n \end{bmatrix}$;    $\Psi^s \leftarrow \mathrm{Wdiag}\left([x_{1s}\, x_{2s} \cdots x_{ns}]\right) \boldsymbol{Z}^T$;    $\Phi^s \leftarrow \boldsymbol{0}$
2   **while** *not converged* **do**
3     $\forall i$, $\boldsymbol{\gamma}_i \leftarrow \exp(\boldsymbol{z}_i^T \Phi^s \boldsymbol{w}_i)$;    $\boldsymbol{D} \leftarrow \boldsymbol{0}$;    $\boldsymbol{r} \leftarrow \boldsymbol{0}$;    $\epsilon = 0.5$;    $\sigma = 10^{-10}$
4     $\nabla g(\Phi^s) \leftarrow -(1/n)(\Psi^s - \boldsymbol{Z}\mathrm{diag}(\boldsymbol{\gamma}) \mathrm{W}^T)$
5     $\mathcal{F} \leftarrow \{(t, j) : t \neq s \text{ and } (|\nabla_{jt} g(\boldsymbol{\phi})| \geq \lambda \text{ or } \phi_{jt} \neq 0 \text{ or } t = 1)\}$
6     **while** *not converged* **do**
7       **for** $(t, j) \in \mathcal{F}$ **do**
8         $a = \sum_{i=1}^n \gamma_i (w_{ij} z_{it})^2$;    $b = \nabla_{jt} g(\Phi^s) + \sum_{i=1}^n \gamma_i w_{ij} z_{it} r_i$;    $c = \phi_{jt} + d_{jt}$
9         $\mu \leftarrow -c + \mathcal{S}_{\lambda/a}(c - b/a)$;    $d_{jt} \leftarrow d_{jt} + \mu$;    $\forall i$, $r_i \leftarrow r_i + \mu z_{it} w_{ij}$
10      **end**
11    **end**
12    **for** $\alpha \leftarrow 1, \epsilon^2, \epsilon^3, \cdots$ **do**
13      $\hat{\Phi}^s \leftarrow \Phi^s + \alpha \boldsymbol{D}$
14      $f(\hat{\Phi}^s) \leftarrow -(1/n)\left(\mathrm{tr}(\Psi^s \hat{\Phi}^s) - \sum_{i=1}^n \exp(\boldsymbol{z}_i^T \hat{\Phi}^s \boldsymbol{w}_i)\right) + \lambda \|\mathrm{vec}(\hat{\Phi}^s)_{\backslash 1}\|_1$
15      **if** $f(\hat{\Phi}^s) \leq f(\Phi^s) + \alpha\sigma[\mathrm{tr}(\nabla g(\Phi^s)^T \boldsymbol{D}) + \|(vec(\Phi^s) + vec(\boldsymbol{D}))_{\backslash 1}\|_1 - \|vec(\Phi^s)_{\backslash 1}\|_1]$ **then**
16        $\Phi^s \leftarrow \hat{\Phi}^s$;    break
17      **end**
18    **end**
19   **end**

---

# E Top 50 Edges for Best LDA and APM Models

Table 3: Top 50 words for LDA (Left) and APM (Right)

| Rank | Evoc. | Edge |
|---|---|---|
| 1 | 38 | woman.n ↔ man.n |
| 2 | 0 | woman.n ↔ wife.n |
| 3 | 13 | train.n ↔ car.n |
| 4 | 69 | school.n ↔ class.n |
| 5 | 0 | drive.v ↔ car.n |
| 6 | 82 | teach.v ↔ school.n |
| 7 | 38 | engine.n ↔ car.n |
| 8 | 35 | publish.v ↔ book.n |
| 9 | 7 | religious.a ↔ church.n |
| 10 | 38 | state.n ↔ government.n |
| 11 | 0 | car.n ↔ bus.n |
| 12 | 32 | year.n ↔ day.n |
| 13 | 25 | seat.n ↔ car.n |
| 14 | 50 | teach.v ↔ student.n |
| 15 | 0 | tell.v ↔ get.v |
| 16 | 38 | wife.n ↔ man.n |
| 17 | 100 | run.v ↔ car.n |
| 18 | 0 | give.v ↔ get.v |
| 19 | 16 | paper.n ↔ book.n |
| 20 | 19 | white.a ↔ black.a |
| 21 | 19 | fish.n ↔ animal.n |
| 22 | 44 | week.n ↔ day.n |
| 23 | 0 | text.n ↔ language.n |
| 24 | 51 | hour.n ↔ day.n |
| 25 | 25 | dog.n ↔ animal.n |
| 26 | 38 | university.n ↔ institution.n |
| 27 | 44 | house.n ↔ government.n |
| 28 | 0 | title.n ↔ subject.n |
| 29 | 13 | ride.v ↔ horse.n |
| 30 | 7 | teacher.n ↔ teach.v |
| 31 | 0 | subject.n ↔ old.a |
| 32 | 19 | west.n ↔ area.n |
| 33 | 7 | people.n ↔ family.n |
| 34 | 16 | tree.n ↔ plant.n |
| 35 | 13 | year.n ↔ week.n |
| 36 | 0 | member.n ↔ authority.n |
| 37 | 0 | high.a ↔ first.a |
| 38 | 0 | urban.a ↔ area.n |
| 39 | 7 | institution.n ↔ date.n |
| 40 | 0 | high.a ↔ area.n |
| 41 | 6 | university.n ↔ date.n |
| 42 | 0 | record.n ↔ play.v |
| 43 | 38 | give.v ↔ church.n |
| 44 | 6 | plant.n ↔ bird.n |
| 45 | 25 | member.n ↔ give.v |
| 46 | 32 | west.n ↔ state.n |
| 47 | 0 | show.v ↔ first.a |
| 48 | 0 | room.n ↔ house.n |
| 49 | 63 | van.n ↔ car.n |
| 50 | 16 | journal.n ↔ book.n |

| Rank | Evoc. | Edge |
|---|---|---|
| 1 | 13 | smoke.v ↔ cigarette.n |
| 2 | 60 | love.v ↔ love.n |
| 3 | 13 | eat.v ↔ food.n |
| 4 | 50 | west.n ↔ east.n |
| 5 | 38 | south.n ↔ north.n |
| 6 | 75 | steel.n ↔ iron.n |
| 7 | 57 | question.n ↔ answer.n |
| 8 | 13 | boil.v ↔ potato.n |
| 9 | 7 | religious.a ↔ church.n |
| 10 | 97 | husband.n ↔ wife.n |
| 11 | 72 | aunt.n ↔ uncle.n |
| 12 | 28 | tea.n ↔ coffee.n |
| 13 | 25 | operational.a ↔ aircraft.n |
| 14 | 0 | competition.n ↔ compete.v |
| 15 | 35 | green.n ↔ green.a |
| 16 | 0 | fox.n ↔ animal.n |
| 17 | 19 | smoke.n ↔ fire.n |
| 18 | 41 | wine.n ↔ drink.v |
| 19 | 33 | troop.n ↔ force.n |
| 20 | 7 | lock.n ↔ key.n |
| 21 | 13 | ride.v ↔ horse.n |
| 22 | 100 | telephone.n ↔ call.n |
| 23 | 76 | politics.n ↔ political.a |
| 24 | 0 | smell.v ↔ smell.n |
| 25 | 7 | teacher.n ↔ teach.v |
| 26 | 7 | check.v ↔ check.n |
| 27 | 72 | printer.n ↔ print.v |
| 28 | 50 | sun.n ↔ earth.n |
| 29 | 0 | rehabilitation.n ↔ contact.n |
| 30 | 35 | salt.n ↔ rice.n |
| 31 | 44 | weekend.n ↔ sunday.n |
| 32 | 35 | publish.v ↔ book.n |
| 33 | 0 | guilty.a ↔ court.n |
| 34 | 35 | copy.v ↔ copy.n |
| 35 | 19 | white.a ↔ black.a |
| 36 | 75 | job.n ↔ employment.n |
| 37 | 75 | room.n ↔ bedroom.n |
| 38 | 38 | morning.n ↔ afternoon.n |
| 39 | 0 | cat.n ↔ animal.n |
| 40 | 0 | similarity.n ↔ sequence.n |
| 41 | 0 | drive.v ↔ car.n |
| 42 | 57 | prison.n ↔ cell.n |
| 43 | 38 | engine.n ↔ car.n |
| 44 | 10 | fall.v ↔ fall.n |
| 45 | 0 | session.n ↔ experience.n |
| 46 | 7 | society.n ↔ class.n |
| 47 | 0 | index.n ↔ close.v |
| 48 | 82 | residential.a ↔ home.n |
| 49 | 51 | mother.n ↔ baby.n |
| 50 | 28 | win.v ↔ prize.n |