[Reviews · NeurIPS 2014]

Submitted by Assigned_Reviewer_10

This paper proposes to learn a text model APM (Inouye+, 2014) for large
datasets by alternating minimization.
APM is an admixture of Poisson random fields on words, thus like an LDA where
topic distributions are replaced by Poisson random fields. As such, learning
possible interactions between words is hard for large vocabularies. Authors
propose an EM-like algorithm where Poisson random field parameters are
optimized in the M step.
The paper also proposes a new metric for evaluating consistency of topic
models, but this is problematic as described below.

Although this paper is clearly written, I cannot find why authors do not use
MCMC for inference. Also while I agree word associations are important for
modeling texts, they are difficult to represent explicitly for huge vocabulary
beyond 1646 in this paper (say, over millions in Google 1-grams).
For modeling such high dimensional discrete data, low-dimensional embeddings
are good solutions and in fact have high performance than LDA, but results
are not compared to such RBM-based text models.

In section 4, authors introduce "evocation" scores that is the sum of
associations between word pairs in the model.
However, this is problematic, because we can assign arbitrarily high scores of
"evocations" with the model by simply assigning high associations for every
word pairs, while giving very low probabilities to the observed data.
In other words, there are no regularization term in the measure and it can
ignore the observed data.
While I agree with the objectives of the authors, such measures cannot be used
to fairly compare between models.

With these considerations, this paper should be more persuasive to show that
Poisson random field text models are in fact superior and learnable for huge
text data we generally encounter.
Summary: Proposes EM-like algorithm for admixture of Poisson random field text models and evaluation metrics.
Lacking comparison to the state-of-the art, and dubious to scale to huge vocabularies.

Submitted by Assigned_Reviewer_17

This paper has two contributions. The first is a new inference scheme
for a compelling topic model and the second is a novel topic model
evaluation metric that explicitly captures whether semantic
relationships are encoded by topic models.

Presentation
============================

This is a very well written paper. Ideas are presented clearly, and
despite the considerable ground to cover, I don't feel that ideas were
shortchagned. In a perfect world, I would have prefered a little more
depth on the model description, but the choices seem reasonable given
the space constraint.

Model
============================

The parallel alternating Newton-like Algorithm for APM allows an
elegant but computationally expensive algorithm to scale to large
datasets. The approach seems to work well and allows application of
the model to a much larger datasets than before.

(This strikes me a similar to the global / local alternations that
have become popular among folks doing stochastic variational
inference.)

Evaluation
============================

My biggest concern is that while the notion of using evocation as a
measure of topic quality seems intuitive, it would be nice to see if
it correlates with human evaluations of topic quality. For instance,
apply the proposed evaluation metric to the Wikipedia, and NYT data
from the Chang et al paper, would we see a positive correlation? This
would be a nice complement to other attempts to automatically evaluate
the coherence of topic models.

This is fairly easy to do, since all of the needed information is on
Boyd-Graber's webpage. When I did this with a quick and dirty script,
only about 300 topics had evocation entries. The correlation of the
evocation score and the topic intrusion precision was about 0.2. The coherence dataset only has a handful of topic words, though, so perhaps the correlation would be better with a deeper set of topics.

Less critically, one of the things that I liked best about the
"Admixture of Poisson MRFs" paper was the argument about discrete word
senses. Given that evocation distinguishes senses, it seems like it
might be possible to have a sense-based evaluation (although it's not
immediately obvious to me how you would do that).
Summary: This paper has two contributions, either of which would make for a good NIPS paper. The first is a new inference scheme
for a compelling topic model and the second is a novel topic model
evaluation metric that explicitly captures whether semantic
relationships are encoded by topic models.

Submitted by Assigned_Reviewer_40

The paper addresses the computational challenge for a recently proposed admixture models of Poisson MRFs to model word dependencies in topic models. The authors proposed a parallelizable Newton-like algorithm which would speed up the computation. Although the paper is technically sound, I do have couple of concerns regarding to the proposed method and experiment results.

- The paper would be much stronger if comparing admixture model with Poisson MRF with some other models. Many other topic models also model the correlation on either document and/or words. For example, hierarchical topic models with nested CRP by Wang et al. models the hierarchies on the topics, hence the correlations among topics and words. Boyd-Graber et al. also proposed topic models for word sense disambiguation using WordNet. Comparison against these models would be a much stronger evidence for the model.

- This paper uses the evocation evaluation metric. It would be great to include some other standard evaluation of correlations, such as PMI (Newman, et al.) of coherence score (Mimno, et al.).

- One other big concern I have regarding to the model is the scalability of the algorithm. The experiments are running on only one dataset, and it is relatively small, e.g., the BNC dataset contains less than 5k documents, with about 1500 word types only. Even though, from the experiment results, it seems like the model took around half an hour to train, after parallelization. It would be great to demonstrate that the model is able to scale to much larger datasets and hence, be potentially more useful to the research community. In addition, the authors conduct experiments only on the BNC dataset, it would be great to include experiments evaluation on other larger datasets to demonstrate the model is efficient and effective.

The paper is considerably well written, and well organized, except couple of comments.

- The paper heavily rely on the knowledge of the recent proposed admixture model of Poisson MRFs, which is a very strong assumption about the readers. Authors should include more explanation about the background knowledge.

- Authors could draw connections with other topic models extensions for correlation modeling.

In terms of significance, the paper address the scalability issue of a recent proposed admixture model of Poisson MRFs, but the experimental results seem less convincing. Author should also focus on the scalability of the model, and examine the algorithm on a much larger dataset to demonstrate the effectiveness and efficiency of the model. Without these, it is hard to see whether the paper would make a significant impact to the research community.

%%%%%%%%%%%%%%%%%%%%%%%%%%%%%%%%%

After check Authors' feedback:

One of my biggest concern is the real scalability of this algorithm. The authors scaled out the inference, but the experiments with only one small dataset are not convincing to show the scalability, which make me doubt how well the proposed algorithm can scale up.

In the author's feedback, they provide a little bit more results with 10k vocab. This is a reasonable size. If they integrate the new results to final version and release code, this is more convincing to me now.

But on the other hand, the evocation evaluation should be better explained and supported, and it would be great if standard evaluation scores like PMI are included as well.
Summary: This paper addresses the scalability issue on a recently proposed admixture model of Poisson MRFs. The experimental results show that the new proposed method yields faster inference procedure and comparable performance with the original algorithm. However, the authors evaluate the method only on one relatively small dataset, which is hard to thoroughly evaluate the effectiveness and efficiency of the proposed method.

Submitted by Assigned_Reviewer_41

The authors scale a new kind of admixture model which allows for word dependence within each topic. Interestingly according to the evaluation metric provided adding more topics does not seem to help. That is admixing the Poisson MRF doesn’t seem that useful and can be harmful. The algorithm presented takes advantage of the substructure of the optimization problem to provided a dramatic increase in scalability. In all exploring admixed PMRFs is worthwhile, and this paper takes a step in trying to understand these models by making them more accessible computationally.

One thing that was unclear to me was if each of the subproblems are convex. Why is there such a big difference between the local solvers? Are they not being run to convergence?
Summary: Exploring admixed PMRFs is worthwhile, and this paper takes a step in trying to understand these models by making them more accessible computationally.
Author Feedback
Author rebuttal: We thank all the reviewers for their helpful comments and would like to primarily address three main concerns:
1. Scalability of algorithm
2. Doubt about evocation metric
3. Comparison to more models

----- Scalability of algorithm -----
As rightly pointed out by the reviewers, the BNC dataset (n=4049 and p=1646) does not properly demonstrate the scalability of the algorithm. Thus, we ran several experiments on Wikipedia (formed by choosing the top 10k words neglecting stop words and then selecting the longest documents). We ran several main iterations of the algorithm with this dataset setting k = 5 and lambda = 0.5:

[n = 20k, p = 5k, density = 18%, # of Words = 50M, nnzTheta = 52k]
First iter: 1 hr, Avg. next three iter: 0.6 hrs
[n = 100k, p = 5k, density = 11%, # of Words = 133M, nnzTheta = 13k]
First iter: 3.1 hrs, Avg. next three iter: 2.2 hrs
[n = 20k, p = 10k, density = 11%, # of Words = 57M, nnzTheta = 57k]
First iter: 3.4 hrs, Avg. next three iter: 2.2 hrs

Note that with n=20k and p=10k, there are 37x more parameters to fit than the BNC dataset. The approximate scaling estimate from this timing is O(n^0.8 p^1.8). Thus, the algorithm is approximately O(np^2), which seems the optimal scaling possible given that there are O(p^2) parameters. We would like to note that even scaling LDA to huge vocabularies has been a difficult and active area of research even many years after the model was introduced (Liu et al. 2011, Tora & Eguchi 2013).

If our paper gets accepted, we will release the code, which to our knowledge would be the first code release for the APM model and would thus be pivotal in researching APM or comparing to it. The code would also provide an excellent timing baseline for future research.

----- Doubt about evocation metric -----
(Reviewer_10) "[A]uthors introduce evocation scores that is the sum of associations between word pairs in the model. However, this is problematic, because we can assign arbitrarily high scores of evocations with the model by simply assigning high associations for every word pairs, while giving very low probabilities to the observed data. In other words, there is no regularization term in the measure, and it can ignore the observed data."

We believe this is a misunderstanding of how we calculate the evocation score. Two options for comparing the top-m pairs are: 1) Pick the top-m pairs based on human evocation scores and then sum the model associations for those pairs. 2) (Our metric) Pick the top-m pairs based on the model associations--note that the ranking is all that matters--and then sum the human evocation scores for those pairs. We agree with Reviewer_10 that (1) would be problematic for the reasons stated but we are actually doing (2). Thus, our evocation metric is similar to the top-m precision for recommender systems in that each topic model "predicts" the top-m word pairs, and this set is evaluated against human judgments of evocation. We apologize if this was unclear from the submitted paper and will make every effort to clarify this point if the paper is accepted.

The above comment by Reviewer_10 encouraged us to rethink the issue of over fitting. Though all of these models are unsupervised (the training algorithms never see the human evocation measurements), there is a small amount of supervision since we select the tuning parameters based on max evocation score. Therefore, we reran the experiments by splitting the human evocation scores into train and test splits (50/50). We selected the model parameters based on the evocation score on the training set and then calculated the final score for comparison on the test split (with m=50 because the train/test splits only have 50% of the original evocation scores). In addition, we trained and evaluated APM for k=50 unlike in the original submission. With this new experimental setup, APM continues to significantly outperform the other models as seen in the results below.

----- Comparison to more models ----
Per the suggestion of the reviewers, we added two more models for comparison: Correlated Topic Models (CTM) and Replicated Softmax (RSM). CTM models correlations between topics (Blei & Lafferty, 2006). RSM is an RBM-based undirected topic model (Hinton & Salakhutdinov, 2009). In addition, we added a random model that randomly ranks all pairs of edges to serve as a naive baseline. We would like to emphasize that all models except APM can only indirectly model dependencies between words through the latent variables since the topic distributions are Multinomial whereas APM can directly model the dependencies between words since the topic distributions are PMRFs. The trends are the same as the submitted paper but here is a sample of the results for all models using the train/test setup described above (in order of increasing score):

[Evoc-1, k=5]
RND: 329, RSM: 356, LDA: 565, CTM: 607, HDP: 701, APM: 1269
[% Improvement over RND]
RND: 0%, RSM: 8%, LDA: 72%, CTM: 85%, HDP: 113%, APM: 286%

The ranking of the models is not always consistent except that APM continues to significantly outperform all other models for k=1,3,5,10 [Evoc-1] and k=1,3,5,10,25,50 [Evoc-2]. Other trends from submitted paper remain the same. We hope that these results with more models and the more fair train/test split evaluation continue to show that APM can be a powerful new topic model for capturing semantically meaningful word pairs.

----- Detailed response for Reviewer_10 -----
"[word associations] are difficult to represent explicitly for huge vocabulary beyond 1646 in this paper"

This would be a reasonable concern if we stored all O(p^2) correlations but we are actually modeling a compact and sparse representation of the joint distribution that only needs O(p) non-zero edges to be stored yet can encode O(p^2) correlations. For example, a chain graph only has (p-1) edges but all the variables are correlated with one another indirectly through the chain.